# Glycolytic reprogramming is involved in tissue remodeling on chronic rhinosinusitis

**Min-Sik Jo[1‡], Hyun-Woo Yang[1‡], Joo-Hoo Park**[1], **Jae-Min Shin**[1,2,3‡]*, **Il-Ho Park**[1,2,3‡]*

**1** Upper Airway Chronic Inflammatory Diseases Laboratory, Korea University College of Medicine, Seoul, South Korea, **2** Medical Device Usability Test Center, Guro Hospital, Korea University College of Medicine, Seoul, South Korea, **3** Department of Otorhinolaryngology-Head and Neck Surgery, Korea University College of Medicine, Seoul, South Korea

‡ MSJ and HWY are co-first authors and JMS and IHP are co-corresponding authors to this work.
* shinjm0601@hanmail.net (JMS); parkil5@korea.ac.kr (IHP)

## Abstract

### Background

Glycolytic reprogramming is a key feature of chronic inflammatory disease. Extracellular matrix (ECM) produced by myofibroblasts plays an important role in tissue remodeling of nasal mucosa in chronic rhinosinusitis (CRS). This study aimed to determine whether glycolytic reprogramming contributes to myofibroblast differentiation and ECM production in nasal fibroblasts.

### Methods

Primary nasal fibroblasts were isolated from the nasal mucosa of patients with CRS. Glycolytic reprogramming was assessed by measuring the extracellular acidification and oxygen consumption rates in nasal fibroblast, with and without transforming growth factor beta 1 (TGF-β1) treatment. Expression of glycolytic enzymes and ECM components was measured by real-time polymerase chain reaction, western blotting, and immunocytochemical staining. Gene set enrichment analysis was performed using whole RNA-sequencing data of nasal mucosa of healthy donors and patients with CRS.

### Result

Glycolysis of nasal fibroblasts stimulated with TGF-B1 was upregulated along with glycolytic enzymes. Hypoxia-inducing factor (HIF)-1α was a high-level regulator of glycolysis, and increased HIF-1α expression promoted glycolysis of nasal fibroblasts, and inhibition of HIF-1α down-regulated myofibroblasts differentiation and ECM production.

**Data Availability Statement:** All relevant data are within the paper and its Supporting information files.

**Funding:** This research was supported by the Bio & Medical Technology Development Program of

the National Research Foundation (NRF), funded by the Korean government (MSIT) (2019M3E5D1A01068992, 2020R1C1C1004572). The funders had no role in study design, data collection and analysis, decision to publish, or preparation of the manuscript.

**Competing interests:** The authors have declared that no competing interests exist.

## Conclusion

This study suggests that inhibition of the glycolytic enzyme and HIF-1α in nasal fibroblasts regulates myofibroblast differentiation and ECM generation associated with nasal mucosa remodeling.

## Introduction

Chronic rhinosinusitis (CRS) is a chronic inflammatory disease that affects 5.5–10% of the population [1]. It is characterized by symptoms, including tenderness, headache, and nasal discharge, which impairs the quality of life of patients. Morphologically, CRS is classified as CRSsNP (without nasal polyps) and CRSwNP (with nasal polyps). However, it is currently classified into Th1, Th2 and Th17 type or eosinophilic, non-eosinophilic CRS based on cytokine expression and immune cells. Among them, some types of CRS strongly accompanied by tissue remodeling on mucosa tend to be recurrent and recalcitrant even after proper management, including medications and surgery [2]. Therefore, our research has focused on tissue remodeling of CRS.

Tissue remodeling refers to the structural changes in damaged tissues due to chronic inflammation or injury, and this process involves continuous extracellular matrix (ECM) production and degradation [3]. Under pathological conditions, uncontrolled ECM production induces irreversible structural changes and leads to recalcitrant and recurrent CRS [4]. ECM accumulation that results from myofibroblasts differentiation and transforming growth factor (TGF)-β1 produced by immune cells and structural cells, is well known as a key activator of this process [5].

Glycolysis and mitochondrial respiration are the two major energy-yielding pathways. Metabolic activity in normal cells depends on mitochondrial oxidative phosphorylation (OXPHOS) to use glucose to generate ATP for energy [6]. However, cells undergoing phenotypic or functional changes have increased glycolysis used for biosynthesis and bioenergetic demand. This phenomenon is referred to as metabolic reprogramming because glycolysis is quicker than OXPHOS in ATP generation [7].

Glycolytic reprogramming in metabolic reprogramming is associated with chronic inflammatory and airway diseases, such as asthma and chronic obstructive pulmonary disease [8–10]. Hypoxia inducible factor (HIF)-1α, a transcription factor, upregulates glycolytic enzymes (hexokinase 2 [HK2], phosphofructokinase-1 [PFK-1], and pyruvate kinase [PK]), which results in glycolytic reprogramming [11]. There is evident that HIF-1α is expressed in fibroblasts of nasal polyps, but it is not known whether this contributes to glycolytic reprogramming [12]. Based on the above background, we hypothesized that glycolytic reprogramming related to HIF-1α contributes to ECM accumulation in CRS. Thus, the purpose of this study was to determine whether glycolytic reprogramming by HIF-1α is involved in myofibroblast differentiation and ECM production in nasal fibroblasts.

## Materials and methods

### Reagents

Human recombinant TGF-β1 was obtained from R&D systems (Minneapolis, MN, USA). The inhibitors, 2-Deoxy-D-glucose (2-DG) for HK2 inhibitor and 3-(3-pyridinyl)-1-(4-pyridinyl)-2-propen-1-one (3-PO) for 6-Phosphofructo-2-Kinase/Fructose-2,6-Biphosphatase (PFKFB) 3

**Table 1. Clinical characteristics of patients (N = 22).**

| Characteristic | UP | CRSsNP-UP | CRSwNP-UP | CRSwNP-NP |
|---|---|---|---|---|
| | (n = 3) | (n = 5) | (n = 6) | (n = 8) |
| No. Women/men | 2/1 | 1/4 | 1/5 | 2/6 |
| Age, y, mean (SD) | 31.7 (1.25) | 46.6 (5.64) | 45.4 (18.83) | 45.2 (16.18) |
| Asthma, no. | 0 | 0 | 0 | 0 |
| Aspirin sensitivity, no | 0 | 0 | 0 | 0 |
| Lund-Mackay CT score, mean (SD) | 0.67 (0.47) | 8 (2.94) | 13.5 (7.95) | 19 (4.23) |

UP, uncinate process; NP, nasal polyp; CRSsNP, chronic rhinosinusitis without nasal polyp; CRSwNP, chronic rhinosinusitis with nasal polyp; SD, standard deviation.

inhibitor, were purchased from Sigma (Burlington, MA, USA). 4-[(2S)-2-amino-2-carbox-yethyl]-N,N-bis(2-chloroethyl) benzeneamine oxide; dihydrochloride (PX-478 2HCl) for HIF-1α inhibitor was purchased from Selleck Chemicals (Houston, TX, USA).

## Human participants and characteristics

All nasal mucosal tissues were obtained from 22 patients (16 males and 6 females; mean age, 43.7 ± 14.9 years). Normal uncinate process tissues (n = 3) were obtained by rhinoplastic surgery. Uncinate process tissues from CRS patients (n = 11) and nasal polyp tissues (n = 8) were obtained from the middle meatus region at the beginning of the endoscopic surgical procedure of CRS patients. The diagnosis of CRS was based on historical, endoscopic, and radiographic criteria, and computed tomography findings of the sinuses according to the 2020 European position paper on rhinosinusitis and nasal polyps (EPOS) guidelines. All participants were recruited from the Department of Otorhinolaryngology, Korea University Medical Center, Korea. None of the patients had taken oral steroids, non-steroidal anti-inflammatory drugs, antihistamines, or antibiotics for at least for months. Informed consent was obtained from all the patients in accordance with the Declaration of Helsinki. This study was approved by the Korea University Medical Center Institutional Review Board (2020GR0308). The clinical characteristics of the patients are summarized in Table 1.

## Isolation and culture of nasal fibroblast

Nasal fibroblasts were obtained from chopped uncinate process tissues of patients who underwent surgery. Uncinate process tissues were isolated by enzymatic digestion with collagenase (500 U/mL; Sigma-Aldrich, St. Louis, MO, USA), hyaluronidase (30 U/mL, Sigma), and DNase (10 U/mL, Sigma). Cells were incubated using Dulbecco's modified Eagle's medium (DMEM, high glucose) with 10% fetal bovine serum (FBS; Invitrogen, Carlsbad, CA), 10,000 mg/mL streptomycin (Invitrogen), and 1% of 10,000 U/mL penicillin (Sigma) in 5% $CO_2$ humidified incubator at 37°C following the supplier's recommendations. After one week of incubation, non-adherent cells were removed by changing the medium. The purity of nasal fibroblasts was confirmed by spindle-shaped cell morphology and flow cytometry.

## Whole transcriptome sequencing

All nasal mucosal tissues (n = 22) were analyzed using whole transcriptome sequencing (RNA-seq) at Macrogen (Seoul, Korea). However, UCSC hg19 was used as a reference for RNA-seq, and a library was constructed using the SureSelectXT RNA Direct reagent kit on the Illumina platform. Sequencing was performed using the NovaSeq 6000 S4 reagent. After a quality control of the raw reads obtained by sequencing, the reads were mapped to the reference genome

using the HISAT2 program. Transcript assembly was performed using the StringTie program with reference-based aligned read information. Data were normalized to fragments per kilobase of transcript per million mapped reads (FPKM) or reads per kilobase of transcript per million mapped reads (TPM). After extracting the expression profile of the normalized data, differential gene expression analysis was performed between normal and CRS patient groups, and gene ontology and Kyoto Encyclopedia of Genes and Genomes analyses were performed.

## Microarray data procedure

Microarray experiments were performed using Illumina HumanHT-12 v3. Biotinylated cRNA was prepared from 0.55 μg total RNA using the Illumina Total Prep RNA amplification kit (Ambion, Austin, TX, USA). Following fragmentation, 0.75 μg was hybridized to the Illumina HumanHT-12 Expression Beadchip according to the protocols provided by the manufacturer. Microarray images were scanned using an Illumina Bead Array Reader confocal scanner. Array data export processing and analysis were performed using the Illumina BeadStudio v3.1.3 (Gene Expression Module v3.3.8). Raw data were normalized using the quantile algorithm.

## Gene set enrichment analysis

Gene set enrichment analysis (GSEA) is a computational method used to determine whether differences in gene sets between two groups are statistically significant. The enrichment score (ES) is the main result of the GSEA, indicating the degree of increase at the top or bottom of the ranked gene list. A positive ES score indicates an increase at the top, and a negative ES score indicates an increase at the bottom. ES normalization was used for comparison with other gene sets. The RNA-sequencing data were analyzed using GSEA v4.2.3. Gene set databases h.all.v.7.5.1 symbols.gmt and NABA_ECM_REGULAOTRS.gmt were used. There were 1000 permutations, and the permutation type was gene set. The data were normalized to the enrichment score to account for the size of the gene set, and a normalized enrichment score (NES) was calculated. It then controls the false positive rate by calculating the corresponding false positive rate (FDR) for each NES. FDR is the estimated probability that a set with a given NES will give a false positive result.

## Glycolysis stress test

Extracellular acidification rate (ECAR) indicates glycolytic rate, which was measured using XF glycolysis stress test kit (Agilent Technologies, Santa Clara, CA, USA). Approximately $1 \times 10^4$ nasal fibroblasts were seeded into each well of XF96 V3 PS cell culture microplate (Agilent Technologies) and incubated at 37°C and 5% $CO_2$. Nasal fibroblasts were pretreated with or without 20 μM PX-478 2HCl (Selleck Chemicals) for 1 h, followed by treatment with or without 5 ng/mL TGF-β1 for 48 h. The ECAR was measured by treating cells with 10 mM glucose, 1 μM oligomycin, and 2-DG 50 mM, sequentially according to the manufacturer's instructions. First, cells are cultured in glycolytic stress test medium without glucose or pyruvic acid and ECAR is measured. The first injection is a saturating concentration of glucose (10 mM), which confirms the maximum of ECAR induced by intracellular glucose. Second, the ATP synthase inhibitor oligomycin is treated. Oligomycin inhibits ATP production and converts energy production into glycolysis. Finally, we process 2-DG, which competitively binds to the first enzyme in the pathway, glucose hexokinase. Then, the corresponding stress was measured through ECAR measurement. Glycolytic capacity is the maximum ECAR rate reached by cells after treatment with oligomycin. It means the value that effectively blocks oxidative phosphorylation and induces the use of glycolysis at its maximum capacity. Glycolytic reserve is a

**Table 2. Sequences of real time-PCR oligonucleotide primers.**

| Primer | | Sequence |
|---|---|---|
| *α-SMA* | Forward | 5'– GTGTTGCCCCTGAAGAGCA–3' |
| | Reverse | 5'– GCTGGGACATTGAAAGTCTCA–3' |
| *Fibronectin* | Forward | 5'– CGGTGGCTGTCAGTCAAAG–3' |
| | Reverse | 5'–AAACCTCGGCTTCCTCCATAA–3' |
| *HIF-1α* | Forward | 5'– GAACGTCGAAAAGAAAAGTCTCG–3' |
| | Reverse | 5'–CCTTATCAAGATGCGAACTCAC–3' |
| *HK1* | Forward | 5'–GCTCTCCGATGAAACTCTCATA–3' |
| | Reverse | 5'–GGACCTTACGAATGTTGGCAA–3' |
| *HK2* | Forward | 5'–GAGCCACCACTCACCCTACT–3' |
| | Reverse | 5'– CCAGGCATTCGGCAATGTG–3' |
| *PFKFB3* | Forward | 5'– TTGGCGTCCCCACAAAAGT–3' |
| | Reverse | 5'– AGTTGTAGGAGCTGTACTGCT –3' |
| *PFKFB4* | Forward | 5'–TCCCCACGGGAATTGACAC–3' |
| | Reverse | 5'–GGGCACACCAATCCAGTTCA–3' |
| *GAPDH* | Forward | 5'–GGAGCGAGATCCCTCCAAAA–3' |
| | Reverse | 5'–GGCTGTTGTCATACTTCTCATG–3' |

measure of how close that function is to the cell's theoretical maximum, and represents the cell's ability to respond to energy demands. Nonglycolytic acidification (basal levels) refers to other extracellular acidification values not attributable to glycolysis.

## Real-time polymerase chain reaction

Total RNA was isolated using RNAiso Plus (TaKaRa Bio, Kusatsu, Shiga, Japan). Complementary DNA (cDNA) was synthesized from 1 μg of RNA templates using M-MLV reverse transcriptase (Promega, Madison, WI, USA), RNase inhibitor (enzymonics, Daejeon, Korea), and oligo dT according to the manufacturer's protocol. Synthesized cDNA was mixed with target primers and Power SYBR Green PCR Master Mix (Applied Biosystems, Foster City, CA, USA) to amplify the target gene, and the amplicons were measured using Quantstudio3 (Applied Biosystems). Sequences of real-time PCR oligonucleotide primers are listed in Table 2. The expression levels of target genes were normalized to target gene / GAPDH ratios. Each experiment was repeated at least three times.

## Western blot analysis

Cells were lysed using RIPA buffer (Cell Signaling Technology, Danvers, MA, USA) for protein extraction. Protein concentrations were quantified using a Pierce™ BCA protein assay kit (Thermo Scientific, Waltham, MA, USA) according to the manufacturer's protocol. A 10-μg protein was mixed with 5X SDS-PAGE loading buffer (Biosesang, Gyeonggi, Korea) and boiled at 95°C for 10 min. Boiled protein samples were separated on 10% SDS-PAGE gel and were transferred to a polyvinylidene difluoride membrane (PVDF; Sigma). Blocking of the PVDF membrane was performed using 5% skimmed milk for 1 h at room temperature. The membrane was washed with TBS-T (Tris-buffered saline with 0.1% Tween 20) three times and then incubated overnight with the primary antibody diluted in 3% bovine serum albumin (BSA) at 4°C. After the overnight incubation, the PVDF membranes were incubated with secondary anti-mouse or anti-rabbit antibodies (Vector Laboratories, Burlingame, CA, USA) at room temperature for 1 h. The blots were detected using the ECL system (Pierce, Rockford, IL,

USA). The primary antibodies used were anti-hypoxia-inducible factor (HIF) -1α (1:200), anti-GAPDH (1:1000), anti-HK1 (1:1000), anti-HK2 (1:1000, Santa Cruz Biotechnology, Dallas, TX, USA), anti-PFKFB3 (1:1000), and anti-PFKFB4 (1:1000, Invitrogen).

## Immunocytochemical staining

Paraformaldehyde (PFA; Biosesang) (4%) was used to fix the cells for 10 min. Triton X-100 (Sigma) (0.2%) in 1% BSA was used to permeabilize the cells for 10 min. Cells were blocked in 3% BSA for 1 h at room temperature and then incubated overnight at 4°C with primary antibody diluted in 3% BSA. The primary antibodies used were anti-HK2 (1:200), anti-PFKFB3 (1:200), anti-PFKFB4 (1:200) and anti-HIF-1α (1:50). Then, the cells were incubated with the secondary antibody for 1 h. The secondary antibodies used were anti-mouse Alexa 488 and anti-rabbit Alexa 555 (1:200, Invitrogen). 4'-6-diamidino-2-phenylindole (DAPI; Invitrogen) was used for nuclear staining for 10 min. The target proteins of the cells were captured and processed using a confocal laser-scanning microscope (LSM 900; Zeiss, Oberkochen, Germany).

## Collagen gel contraction assay

Rat-tail tendon collagen type 1 (Discovery Labware Inc., Bedford, MA, USA) was mixed with $3 \times 10^5$ nasal fibroblasts in serum-free DMEM. Collagen mixture (500 μL), containing fibroblasts were dispensed into each well of a 24-well cell culture plate to solidify the gel and incubated at 37°C for 2 h. Serum-free DMEM (500 μL) DMEM with 5 ng/mL TGF-β1 or 20 nM PX-478 was added to each well. The gels were incubated at 37°C in 5% $CO_2$ for 72 h. The surface area of the collagen gels was quantified using the ImageJ software (NIH).

## Sircol soluble assay

Soluble collagen assay Sircol™ was purchased from Biocolor Ltd. (Belfast, U.K.). The spent medium of fibroblasts cultured at 37°C in 5% $CO_2$ for 72 h was collected. The spent medium was centrifuged to obtain the supernatant. The dye reagent (1,000 μL) was added to 100 μL of the supernatant, and the mixture was incubated at room temperature for 30 min. Further, the samples were centrifuged for 30 min, and the supernatant was discarded, washed with acid-salt wash reagent, and centrifuged. The pellet was dissolved in 250 μL alkaline reagent, and 200 μL samples were transferred to each well of a 96-well plate and measured using a fluorescence microplate reader.

## Statistical analysis

The results were obtained from three independent experiments. Data are expressed as mean ± standard deviation, and an analysis of variance test was used to compare two or more groups. A t-test was performed for comparison of the two groups. Analyses were performed using GraphPad Prism 5 (GraphPad Software 8.0; San Diego, CA, USA). Statistical significance was set at $p < 0.05$.

## Results

### Enhanced glycolytic flux, HIF-α, and glycolytic enzymes in TGF-β1-treated nasal fibroblasts

Glycolytic flux is the rate of ATP production and is regulated by the rate-limiting enzymes HK2, as well as indirect enzymes, PFKFB3 and PFKFB4. To investigate whether glycolytic flux and the expression of HIF-1α and glycolytic enzymes are elevated by TGF-β1, nasal fibroblasts

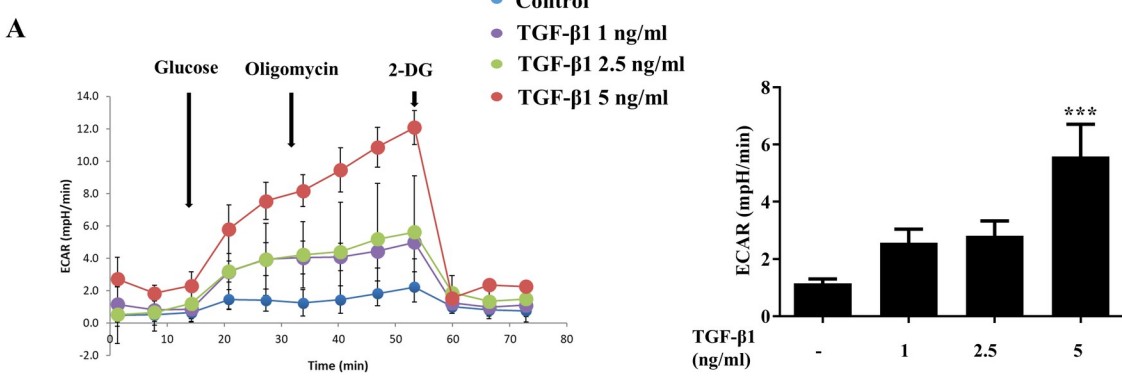

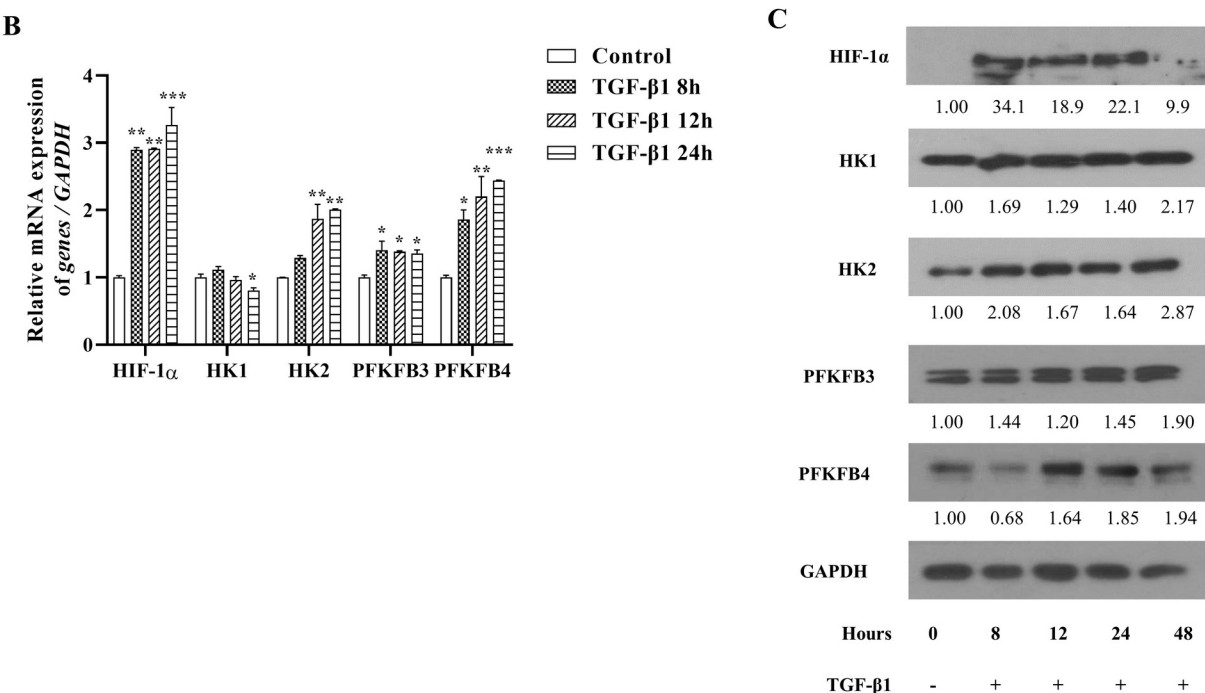

**Fig 1. Glycolytic rate, hypoxia inducible factor (HIF)-1α, and glycolytic enzymes were upregulated in transforming growth factor (TGF)-β1-treated nasal fibroblast.** (a) Extracellular acidification rate (ECAR) of fibroblast treated with or without TGF-β1 was measured using XFe96; (b) Gene expression of HIF-1α, hexokinase (HK) 1, HK2, 6-phosphofructo-2-kinase/fructose-2,6-bisphosphatase (PFKFB) 3, and PFKFB4 were confirmed by RT-PCR; and (c) protein levels were detected by Western blotting. $^{*}p < 0.05$, $^{**}p < 0.01$, $^{***}p < 0.001$ compared with the control using one-way ANOVA test.

were examined using the XF glycolysis stress test, RT-PCR, and western blotting. The ECAR significantly increased after treatment with 5 ng/mL of TGF-β1 (Fig 1a). The gene expression and protein levels of HIF-1α, HK2, PFKFB3, and PFKFB4 were significantly elevated by TGF-β1 (5 ng/ml). However, the mRNA level of HK1 was decreased at 24 h, and the protein level of HK1 did not change (Fig 2b and 2c). These results indicate that a glycolysis regulator (HIF-1α), rate-limiting direct (HK2), and indirect enzymes (PFKFB3 and PFKFB4) were involved in TGF-β1-induced glycolysis in nasal fibroblasts, but HK1 was not associated with this process.

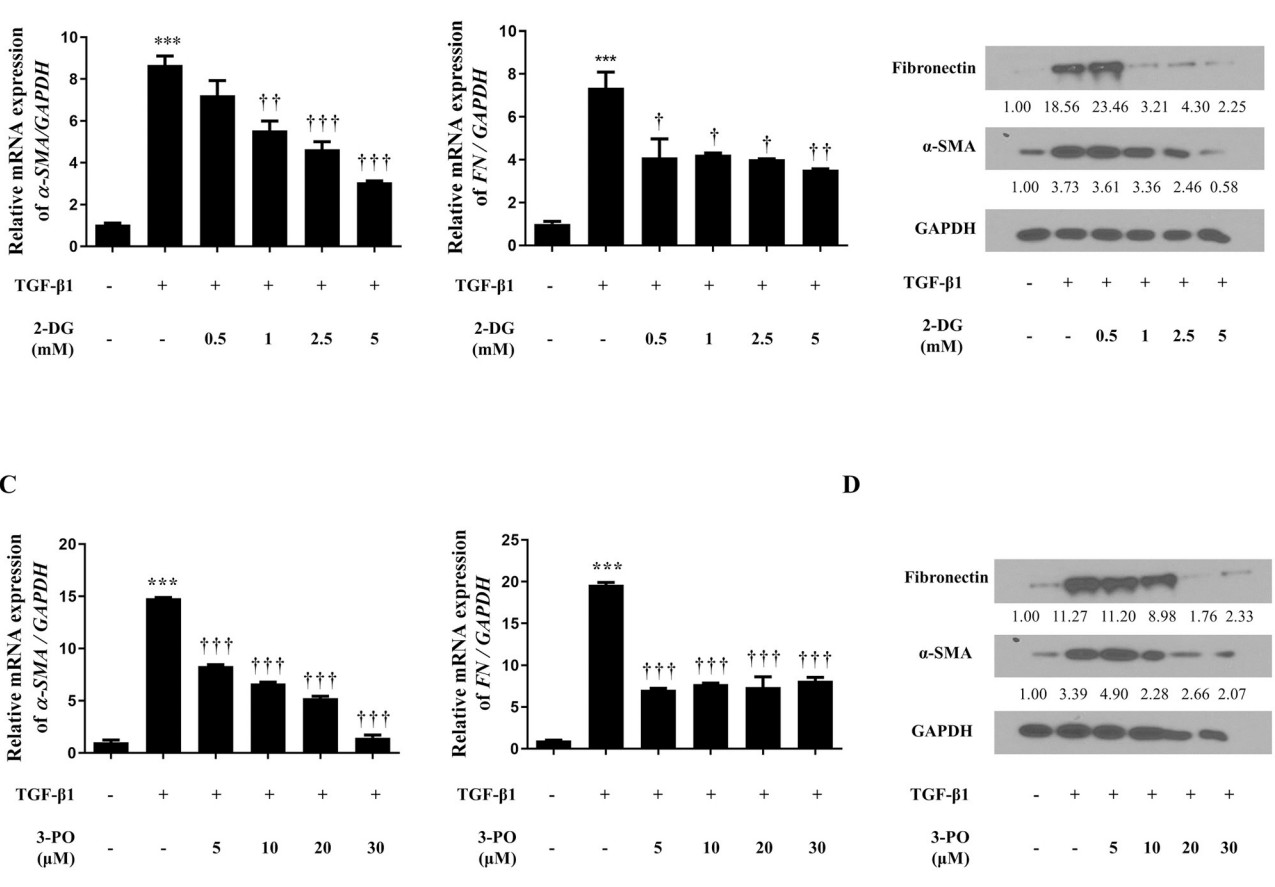

**Fig 2. 2-Deoxy-D-glucose (2-DG) and 3-(3-pyridinyl)-1-(4-pyridinyl)-2-propen-1-one (3-PO) downregulated α-SMA (SMA) and fibronectin expression in TGF-β1-treated nasal fibroblasts.** (a, c) Gene expression of α-SMA and fibronectin was confirmed by RT-PCR; and (b, d) protein levels were detected by Western blotting. ***p < 0.001 compared with control by one-way ANOVA test, †p < 0.05, ††p < 0.01, †††p < 0.001 compared with TGF-β1 using one-way ANOVA test.

## Inhibition of glycolytic enzymes downregulates myofibroblast differentiation and ECM production in TGF-β1-treated nasal fibroblast

In a previous experiment, it was confirmed that glycolysis was upregulated in nasal fibroblasts treated with TGF-B1, and TGF-B1 induced overexpression of glycolytic enzymes HK2 and PFKFB3. Therefore, inhibition experiment was performed to confirm whether modulation of glycolytic enzymes (HK2, PFKFB3) depressed the myofibroblast differentiation and ECM of TGF-B1-stimulated nasal fibroblasts. Nasal fibroblasts were pretreated with 2-DG (HK2 inhibitor; 0.5 to 5 mM) and 3-PO (PFKFB3 inhibitor; 5–30 μM) for 1 h and stimulated with TGF-β1 (5 ng/ml) for 24 h. The levels of RNA and protein of α-SMA and fibronectin increased by TGF-β1 treatment. The treatment of 2-DG or 3-PO were significantly downregulated the tissue remodeling related markers (Fig 2a–2d). These results suggest that HK2 and PFKFB3 are related to myofibroblast differentiation and ECM production through glycolysis in nasal fibroblasts.

### PX-478 downregulates glycolytic enzymes and glycolytic flux in TGF-β1-treated nasal fibroblasts

To examine whether inhibition of HIF-1α decreases glycolytic enzymes and glycolytic flux, HIF-1α inhibition experiments were performed. Nasal fibroblasts were pretreated with PX-478 (20 nM); specific HIF-1α inhibitor. The gene expression of HIF-1α, HK2, PFKFB3, and PFKFB4 was significantly decreased by PX-478 (Fig 3a). The protein levels of HIF-1α, HK2, PFKFB3, and PFKFB4 were reduced by PX-478 in TGF-β1-treated nasal fibroblasts (Fig 3b and 3c). The ECAR was significantly decreased in nasal fibroblasts treated with PX-478 compared to that in TGF-β1-only fibroblasts (Fig 3d). These results show that TGF-β1-induced *HIF-1α* expression was decreased by PX-478 treatment. Additionally, PX-478 downregulated glycolytic enzymes and glycolytic flux in TGF-β1-treated nasal fibroblast. Glycolysis was upregulated by TGF-β1 and downregulated by PX-478. These results suggested that TGF-β1 induces glycolytic reprogramming through HIF-1α-regulated glycolytic enzymes in nasal fibroblasts. It also suggests that HIF-1α may act as an upstream regulator of glycolysis.

### Inhibition of TGF-β1-induced HIF-1α expression decreases myofibroblast differentiation and ECM production in nasal fibroblasts

To determine whether the regulation of glycolysis through inhibition of HIF-1a can regulate myofibroblast differentiation and ECM generation, an inhibition experiment with PX-478 was performed. TGF-β1 significantly increased α-SMA and fibronectin expression. On the other hand, PX-478 decreased the expression of α-SMA and fibronectin expression in fibroblasts by inhibiting the expression of HIF-1a and glycolytic enzymes induced by TGF-β1. (Fig 4a–4c). Total collagen protein was evaluated by Sircol soluble collagen assay and was significantly increased by TGF-β1 and decreased by PX-478 (Fig 4d). The contractile ability of myofibroblasts was measured using collagen gel contraction assay. PX-478 inhibited collagen gel contraction compared with that in nasal fibroblasts treated with TGF-β1-only (Fig 4e). These results showed that inhibition of TGF-β1-induced HIF-1α downregulated the fibroblast-to-myofibroblast transition, ECM accumulation, and myofibroblast function, such as contractile ability in TGF-β1-treated nasal fibroblasts.

### Glycolysis and ECM gene sets were increased in CRS patients and TGF-β1 treated nasal fibroblasts

To confirm the changes in the biological processes of nasal mucosa tissue and fibroblasts, transcripts obtained from nasal mucosa tissues and fibroblasts were analyzed using GSEA. In the CRS patients, GSEA showed a different biological status compared to the control, and gene sets of glycolysis and ECM regulators were listed in the top 20 positively enriched gene sets (Fig 5a). The NES of glycolysis was 1.509 (p = 0.0337), and the NES of ECM regulators was 1.659 (p = 0.0281), both of which are significant in the CRS patients. The GSEA results for TGF-β1-treated nasal fibroblasts are also listed. The NES of glycolysis (1.7861153, p < 0.001) and ECM regulators (1.3441843, p = 0.04619) were significant in the nasal fibroblasts. These results suggest that an increased biological status, including glycolysis and ECM is involved in the pathogenesis of CRS. Additionally, the altered biological status of TGF-β1-treated nasal fibroblasts suggests that glycolysis is involved in ECM accumulation in nasal myofibroblasts.

## Discussion

In this study, we demonstrated that increased glycolysis through HIF-1α contributes to myofibroblast differentiation and ECM generation in nasal fibroblasts. TGF-β1-treated nasal

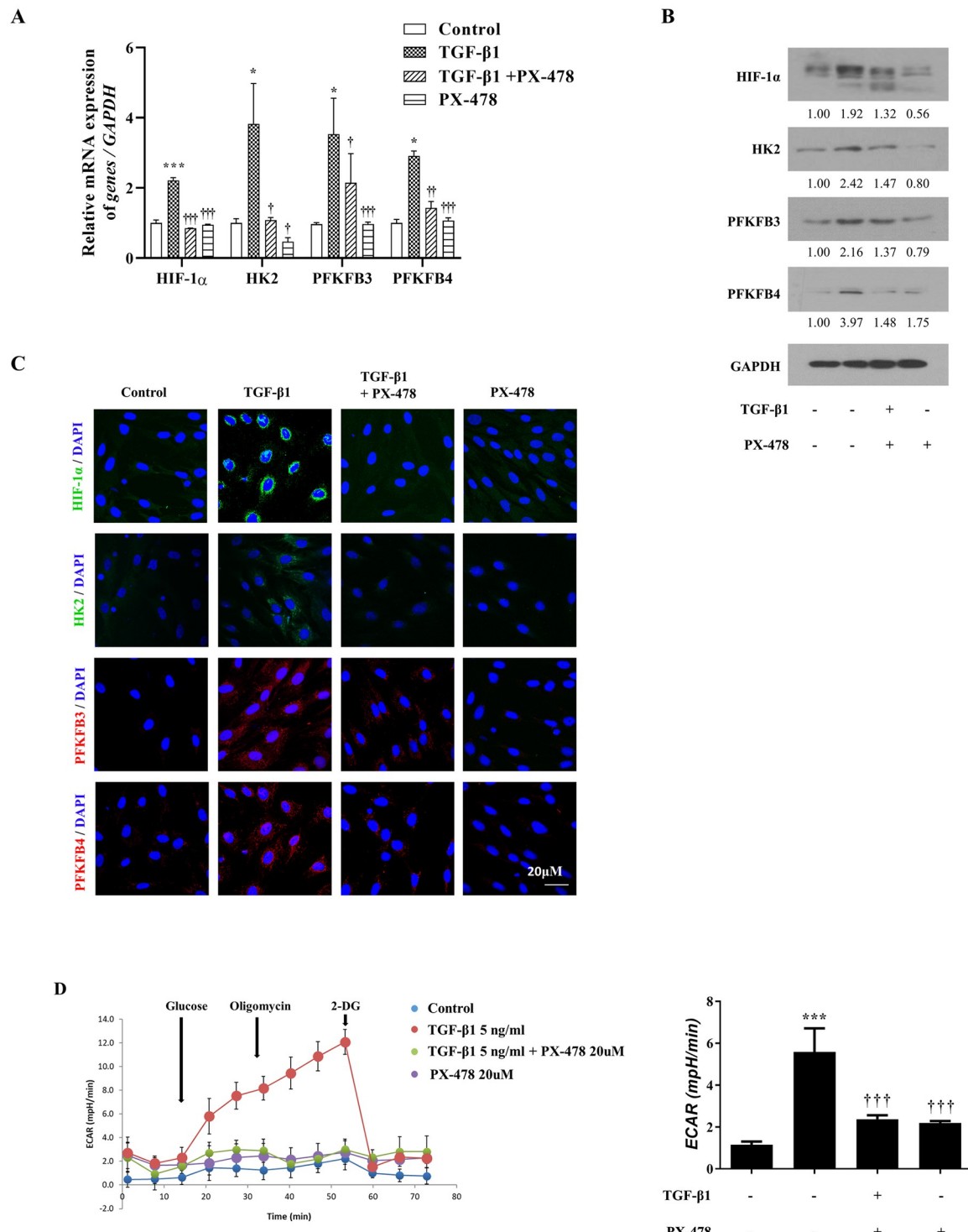

**Fig 3. PX-478 decreased the expression of HIF-1α and glycolytic enzymes, and downregulated glycolytic flux in TGF-β1-treated nasal fibroblast.** (a) Gene expression levels of HIF-1α, HK2, PFKFB3, and PFKFB4 were confirmed by RT-PCR; and (b) Protein levels were detected by Western blotting; (c) Protein expression and localization of HIF-1α, HK2 (Green), PFKFB3, PFKFB4 (red), and DAPI (blue) were determined by immunofluorescence staining; (d) ECAR was measured using XFe96. $^*p < 0.05$ and $^{***}p < 0.001$ compared with control using one-way ANOVA test, $†p < 0.05$, $††p < 0.01$, and $†††p < 0.001$ compared with TGF-β1 using the one-way ANOVA test.

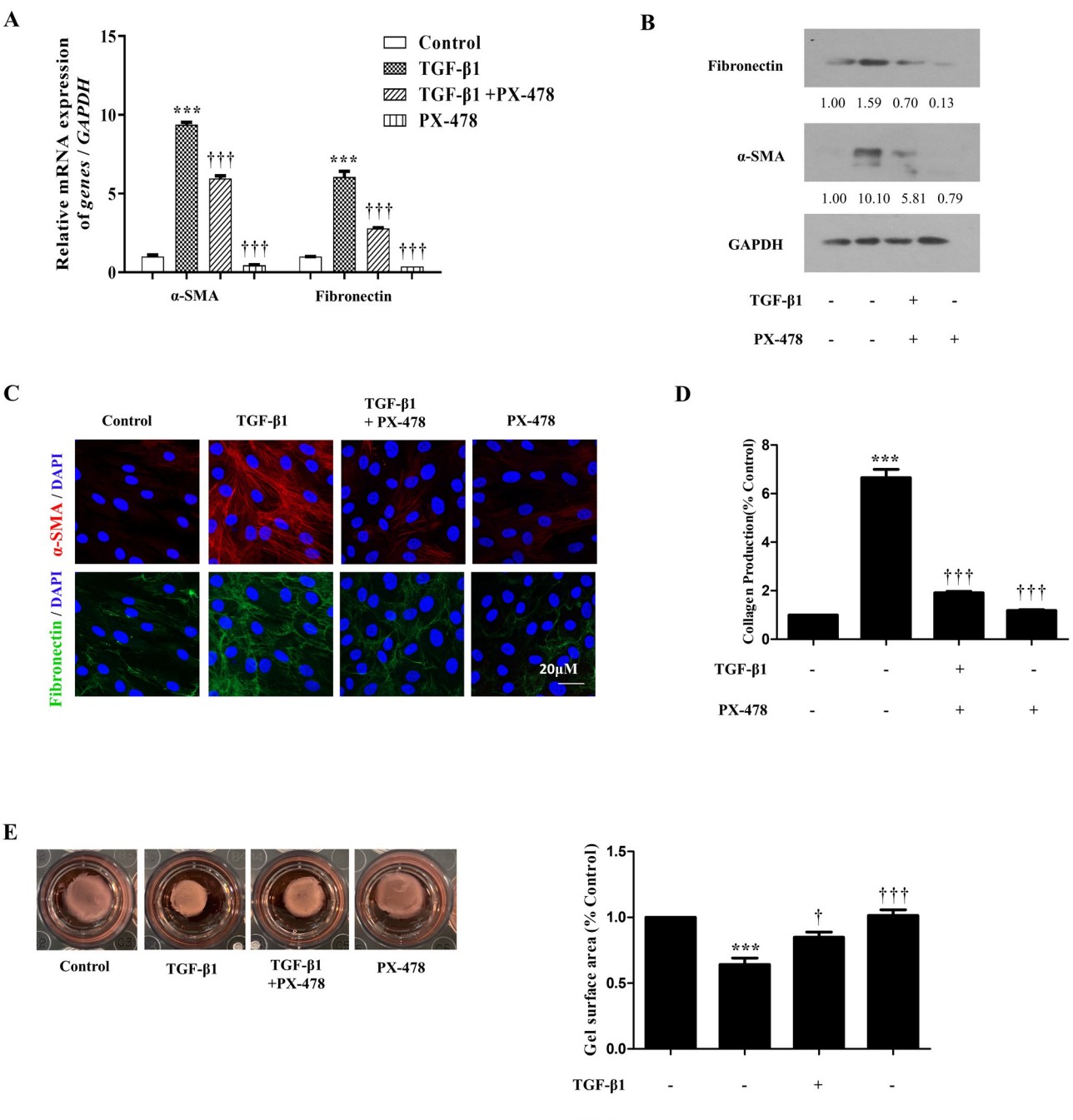

**Fig 4. PX-478 inhibited expression of α-SMA, fibronectin, and collagen.** (a) Gene expressions of α-SMA and fibronectin were confirmed by RT-PCR; and (b) Protein levels were detected by Western blotting. (c) Immunofluorescence was detected using a confocal laser scanning microscope. α-SMA (red), fibronectin (green), and DAPI (blue); (d) Total collagen production was measured using Sircol soluble collagen assay; (e) Contractile activity was measured using a collagen gel contraction assay, and the collagen gel areas were measured using an Image J analyzer. ***p < 0.001 compared with control by one-way ANOVA test, †p < 0.05, †††p < 0.001 compared with TGF-β1 by one-way ANOVA test.

fibroblasts up-regulate the expression of glycolysis and glycolytic enzymes. Also, this glycolytic reprogramming causes tissue remodeling of nasal mucosa by upregulating α-SMA and fibronectin. It was confirmed that these pathways are regulated by HIF-1a or glycolytic enzyme inhibitors. PX-478, a HIF-1α specific inhibitor, inhibited the production of glycolytic enzymes,

**A**

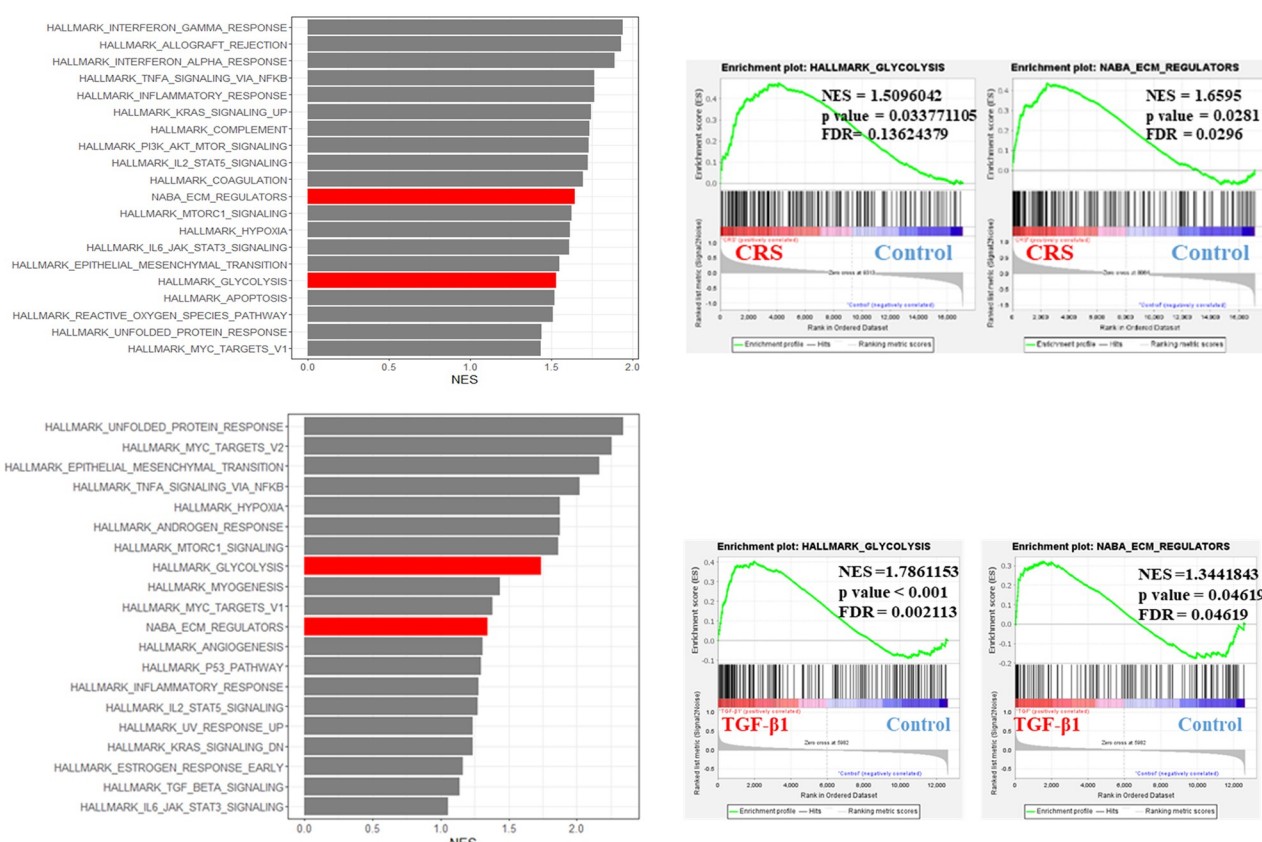

**Fig 5. Gene set enrichment analysis (GSEA) were analyzed using transcripts of chronic rhinosinusitis (CRS) patients and TGF-β1-treated nasal fibroblast.** (a) Normalized enrichment scores (NES) of the top 20 positively enriched gene sets, red is glycolysis and ECM REGULATORS gene sets in CRS patients. NES, p-value and false discovery rate (FDR) of glycolysis gene set and ECM REGULATORS gene set in CRS patients. NES of the top 20 positively enriched gene sets in TGF-β1-treated nasal fibroblast. NES, p-value and FDR of glycolysis gene set and ECM REGULATORS gene set in TGF-β1-treated nasal fibroblast.

α-SMA, fibronectin, collagen, and collagen contraction in TGF-β1-treated nasal fibroblasts. These findings indicate that HIF-1α-regulated glycolysis contributes to TGF-β1-induced myo-fibroblast differentiation and ECM deposition in nasal fibroblasts. Inflammation is a physio-logical defense mechanism triggered by harmful external stimuli, such as infection, injury, and pathogens. Timely immune responses effectively eliminate harmful stimuli and are essential for the restoration of tissue homeostasis. Unresolved inflammation caused by a dysregulated immune response leads to chronic inflammatory disease, which causes irreversible structural changes in damaged tissue [13, 14]. Although irreversible structural changes have been recog-nized as a result of persistent inflammation, recent research has shown that inflammation and tissue remodeling occur simultaneously after injury [15]. Therefore, tissue remodeling is as important as immune response during the inflammatory process. The ECM plays a key role in tissue stability and elasticity, and is a complex three-dimensional structure composed of colla-gen, glycoprotein, and fibronectin [16]. In the pathology of chronic inflammatory diseases, ECM deposition is associated with immune response by activating, differentiating, and surviv-ing immune cells [3]. Tissue remodeling, such as ECM accumulation in lower airway diseases,

including chronic obstructive pulmonary disease and idiopathic pulmonary fibrosis, contributes to disease prognosis [17]. In the present study, we investigated the differential expression of transcripts between CRS patients and healthy controls to identify gene sets that constitute the biological processes involved in the pathogenesis of CRS. Our data showed that altered biological processes in nasal mucosa tissue of CRS patients were analyzed using GSEA, and gene sets of ECM regulators were upregulated in nasal mucosa tissues. This phenomenon was also observed in TGF-β1-treated nasal fibroblasts. Meng et al. suggested that the expression of α-SMA and ECM was increased in early-stage polyps, and excessive ECM deposition by nasal myofibroblasts contributes to tissue remodeling in CRS [18]. This supports our results that dysregulation of the ECM is involved in tissue remodeling associated with the pathological cause of CRS, and that nasal myofibroblasts expressing α-SMA contribute to ECM accumulation in the nasal mucosa.

Glycolysis, one of the metabolic pathways, breaks down glucose into 2 pyruvate and 2 ATP molecules in 10 steps. It is transported to the mitochondria and is ultimately converted to 32 ATP and $CO_2$ via oxidative phosphorylation. Glycolysis is less efficient than oxidative phosphorylation, but is preferred for proliferating cells because of its high rate of ATP generation [6]. Metabolic reprogramming occurs under pathological conditions including inflammation. Recent research has indicated that glycolytic reprogramming is associated with the pathogenesis of inflammatory diseases and dysregulated glycolysis by enhanced glycolytic enzymes is considered a therapeutic target [19]. It has been reported that 2-DG and 3-PO inhibit glycolysis, inhibit the activation of immune cells through excessive glycolysis, and reduce the immune response in inflammatory diseases [20]. In a previous study, collagen deposition was induced by enhanced glycolysis through HK2 and PFKFB3 in lung fibroblasts [21, 22]. Our results demonstrated that glycolysis was enhanced by TGF-β1, and this process involves glycolytic enzymes, such as HK2, PFKFB3, and PFKFB4; however, HK1 was not involved in enhanced glycolysis in nasal fibroblasts. Yin et al. found that HK2 was upregulated, but HK1 did not change in human lung fibroblasts[22]. However, Wang et al. suggested that the expression of HK1 was increased by equal doses of TGF-β1, indicating that HK1 is a key glycolytic enzyme in rat lung fibroblasts [23]. It is reasonable to hypothesize that the difference in species between humans and rats may induce the expression of HK1 in primary fibroblasts. In the present study, HK2 and PFKFB3, induced by TGF-β1, contributed to the expression of α-SMA and fibronectin in nasal fibroblasts. These findings indicate that TGF-β1 secreted under inflammatory conditions induces glycolytic reprogramming by upregulating glycolytic enzymes, which contributes to myofibroblast differentiation and ECM production in nasal fibroblasts. Thus, glycolytic reprogramming is involved in the ECM accumulation produced by nasal myofibroblasts related to CRS tissue remodeling.

Hypoxia-inducible Factor (HIF)-1 is a transcription factor that upregulates glycolytic metabolism [24]. It is a dimeric protein complex composed of HIF-1α and HIF-1β and is regulated by the degradation of HIF-1α in the presence of oxygen. However, recent studies have suggested that HIF-1α is stabilized by TGF-β1 in the presence of oxygen and is implicated in ECM accumulation in lower airway disease [25, 26]. Based on previous studies, HIF-1α is considered a new therapeutic target for uncontrolled ECM synthesis [27]. Used as an inhibitor of HIF-1α, PX-478 is a small-molecule compound with anti-tumor activity. Koh et al. determined that HIF-1α expression is inhibited by PX-478 in epithelial cell lines. Furthermore, the mechanism that may account for the reduction in HIF-1α protein by PX-478 is primarily through inhibition of HIF-1α translation, and to a lesser extent, decreasing levels of HIF-1α mRNA and inhibiting HIF-1α deubiquitination [28]. Villa-Roel et al. previously indicated that PX-478 reduces ECM by inhibiting the expression of HIF-1α in human aortic endothelial cells [29].

Here, we confirmed that the expression of HIF-1α in nasal fibroblasts was induced by TGF-β1 and decreased by PX-478. Our results showed that PX-478 not only inhibited by glycolytic enzymes (HK2, PFKFB3, and PFKFB4) but also reduced α-SMA, fibronectin, and collagen production in nasal fibroblasts. Moreover, PX-478 significantly reduced ECAR, but OCR was not significantly downregulated (S1 Fig). These results suggest that glycolytic metabolism is shifted by HIF-1α-regulated glycolytic enzymes in nasal fibroblasts, and stabilization of HIF-1α contributes to myofibroblast differentiation and ECM accumulation related to tissue remodeling in CRS. Additionally, we confirmed that TGF-β1 increased mitochondrial function and glycolysis in nasal fibroblasts, suggesting that mitochondrial respiration is also associated with phenotypic and functional changes in nasal fibroblasts. Negmadjanov et al. suggested that TGF-β1-induced mitochondrial function is involved in myofibroblast differentiation and ECM production in mouse fibroblast [30]. However, it is still unclear whether changes in mitochondrial function of nasal fibroblasts contribute to pathologic conditions should be studied in the future. To our knowledge, this is the first study to show that glycolytic reprogramming regulated by HIF-1α is associated with myofibroblast differentiation and ECM production in TGF-β1-treated nasal fibroblasts.

Additionally, reprogramming of glucose metabolism in epithelial cells has also been reported to contribute to CRS. Zheng et al. reported that proinflammatory cytokines stimulate epithelial cells to upregulate glycolysis in CRS. This increased glycolysis induces the expression of proinflammtory cytokine in nasal epithelial cells and aggravates CRS through positive feedback [31]. Structural cells such as fibroblasts and epithelial cells present in the musoca not only play a role in tissue morphology and primary defense, but also act as immune regulators by responding to external antigens or cytokines. Therefore, we plan to conduct a study on whether glycolysis is involved in the immunomodulatory response of these structural cells in the future. In addition, CRS will be classified into subtypes according to immune responses, and glycolysis reprogramming in each type will be studied.

In conclusion, our study demonstrated that glycolytic shift through glycolytic enzymes contributes to myofibroblast differentiation and ECM production in nasal fibroblasts, and that HIF-1α is a regulator of glycolysis in nasal fibroblasts. Furthermore, the mechanisms of metabolic reprogramming are important areas for further study, and additional research is needed to understand tissue remodeling related to CRS pathogenesis.

## Supporting information

**S1 Raw images.**
(PDF)

**S1 Fig. Measurement of mitochondrial function in nasal fibroblasts with and without TGF-β1 or with and without PX-478 To investigate whether mitochondrial function are elevated by TGF-β1, nasal fibroblast was examined using XF cell mitostress test.** Oxygen consumption rate (OCR) was significantly increased at 5 ng/ml TGF-β1 (S1A Fig). To determined whether PX-478 decrease mitochondrial function, PX-478 (20 nM) was pretreated for 1 hour, followed by TGF-β1 48 hours. PX-478 downregulated OCR compared to TGF-β1- only, but it wasn't significantly (S1B Fig). FCCP: Carbonyl cyanide 4-(trifluoromethoxy) phenylhydrazone. Oligomycin affects electron flow through the electron transport chain (ETC), reducing the mitochondrial respiration level. FCCP is an uncoupling agent that disrupts the proton gradient and the mitochondrial membrane potential. As a result, electron flow through the ETC is not inhibited and oxygen consumption by complex IV reaches a maximum. The last is a mixture of Rotenone, a Complex I inhibitor, and Antimycin A, a Complex III inhibitor. This combination blocks mitochondrial respiration and

enables the calculation of non-mitochondrial respiration driven by processes outside the mitochondria.
(TIF)

## Author Contributions

**Conceptualization:** Min-Sik Jo, Jae-Min Shin, Il-Ho Park.

**Data curation:** Min-Sik Jo, Joo-Hoo Park.

**Formal analysis:** Min-Sik Jo, Joo-Hoo Park.

**Writing – original draft:** Min-Sik Jo, Jae-Min Shin.

**Writing – review & editing:** Hyun-Woo Yang, Jae-Min Shin, Il-Ho Park.

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
