## [Decision Letter · Decision Letter 0]

3 Jan 2023

PONE-D-22-33119Glycolytic reprogramming is involved in tissue remodeling on chronic rhinosinusitisPLOS ONE

Dear Dr. Jae-Min Shin

Thank you for submitting your manuscript to PLOS ONE. After careful consideration, we feel that it has merit but does not fully meet PLOS ONE’s publication criteria as it currently stands. Therefore, we invite you to submit a revised version of the manuscript that addresses the points raised during the review process.

We look forward to receiving your revised manuscript.

Kind regards,

Kishor Pant

Academic Editor

PLOS ONE

Manuscript Number:

“This research was supported by the Bio & Medical Technology Development Program of the National Research Foundation (NRF), funded by the Korean government (MSIT)  (2019M3E5D1A01068992, 2020R1C1C1004572)”

“This research was supported by the Bio & Medical Technology Development Program of the National Research Foundation (NRF), funded by the Korean government (MSIT) (2019M3E5D1A01068992).”

“This research was supported by the Bio & Medical Technology Development Program of the National Research Foundation (NRF), funded by the Korean government (MSIT)  (2019M3E5D1A01068992, 2020R1C1C1004572)”

“None”

7. Please include your tables as part of your main manuscript and remove the individual files. Please note that supplementary tables (should remain/ be uploaded) as separate "supporting information" files

Additional Editor Comments:

Dear Dr. Jae-Min Shin, Thank you again for your manuscript submission. Your manuscript has been reviewed by experts in the field.

1-Please check and update your manuscript in accordance to the referees' recommendations and submit the revised files.

2-The authors also need to improve the quality of the figures.

3-Send the editors and referees a brief cover letter explaining your modifications.

Reviewers' comments:

Reviewer's Responses to Questions

**Comments to the Author**

1. Is the manuscript technically sound, and do the data support the conclusions?

Reviewer #1: Partly

Reviewer #2: Yes

2. Has the statistical analysis been performed appropriately and rigorously? 

Reviewer #1: Yes

Reviewer #2: Yes

3. Have the authors made all data underlying the findings in their manuscript fully available?

Reviewer #1: Yes

Reviewer #2: Yes

4. Is the manuscript presented in an intelligible fashion and written in standard English?

Reviewer #1: Yes

Reviewer #2: Yes

5. Review Comments to the Author

Reviewer #1: General comments:

In this study, Min-Sik Jo and his colleagues studied the relationship between glycolytic reprogramming and tissue remodeling in chronic rhinosinusitis. They found that glycolysis of nasal fibroblasts under the TGF-B1 stimulation was upregulated along with glycolytic enzymes, and inhibition of HIF-1a downregulated myofibroblasts differentiation and ECM production. There are major concerns that should be addressed seeing as follows.

Major comments:

1. All Western blotting bands need to be quantitatively analyzed. In addition, the bands of GAPDH shown in Figure 1C are connected, indicating that the loading amount of the protein was too high.

2. What does ECAR refer to in all the figures, basic glycolysis, glycolysis capacity, or glycolysis reserve?

3. In Figure 3D, TGFβ group and TGFβ+PX-478 group had no significant increase of ECAR after giving glucose. Please explain it.

4. The authors stated that they studied the role of glycolytic reprogram in CRS. However, they only focused on glycolysis, and provided no data regarding oxidative phosphorylation.

5. The authors should compare the difference between CRSsNP and CRSwNP.

6. Chen et al recently studied glycolysis in CRS epithelial cells and its involvement in the pathogenesis of CRS (https://doi.org/10.1016/j.jaci.2022.09.036). This recent study should be compared and discussed.

7.

Minor comment:

1. On line 354, the format of the reference was different from the others.

Reviewer #2: Manuscript titled “Glycolytic reprogramming is involved in tissue remodeling on chronic rhinosinusitis” Overall, the manuscript is well drafted and the data that was generated supports the conclusion presented by the authors. However, some questions need to be addressed by the author to further improve the manuscript.

1. Did the authors study any cytokines both proinflammatory and anti-inflammatory.

3. In ECAR experiment did the authors had done normalization of data. If not please include in revised manuscript

3.The figures are not clear enough to read. Please include high quality figures with font size that is clearly visible and readable.

6. PLOS authors have the option to publish the peer review history of their article (what does this mean?). If published, this will include your full peer review and any attached files.

Reviewer #1: No

Reviewer #2: **Yes: **jasvinder singh

<quillbot-extension-portal></quillbot-extension-portal>

---

## [Author Response · Author response to Decision Letter 0]

18 Jan 2023

Dear Editor

Thank you for sending us many good comments about the study. We sincerely responded to the opinions of the reviewer. I've attached a response to the comments below.

Review #1

Major comments:

1. All Western blotting bands need to be quantitatively analyzed. In addition, the bands of GAPDH shown in Figure 1C are connected, indicating that the loading amount of the protein was too high.

- Thank you for your comments. We quantified all western blot data and added them to the figures. GAPDH data have been corrected.

2. What does ECAR refer to in all the figures, basic glycolysis, glycolysis capacity, or glycolysis reserve?

- Glycolysis is the process by which glucose is converted to pyruvate and ATP is produced. The XF Glycolysis Stress Test presents the measure of glycolysis as the ECAR rate reached by a given cell after the addition of saturating amounts of glucose. The first injection is a saturating concentration of glucose. The cells utilize the glucose injection and catabolize it through the glycolytic pathway to pyruvate, producing ATP, NADH, water, and protons. The extrusion of protons into the surrounding medium causes a rapid increase in ECAR. This glucose-induced response is reported as the rate of glycolysis under basal conditions. The second injection is oligomycin, an ATP synthase inhibitor. Oligomycin inhibits mitochondrial ATP production, and shifts the energy production to glycolysis, with the subsequent increase in ECAR revealing the cellular maximum glycolytic capacity. The final injection is 2-deoxy-glucose (2-DG), a glucose analog that inhibits glycolysis through competitive binding to glucose hexokinase, the first enzyme in the glycolytic pathway. The resulting decrease in ECAR confirms that the ECAR produced in the experiment is due to glycolysis. The difference between glycolytic capacity and glycolysis rate defines glycolytic reserve. ECAR, prior to glucose injection, is referred to as nonglycolytic acidification; caused by processes in the cell other than glycolysis.

• Glycolytic capacity: This measurement is the maximum ECAR rate reached by a cell following the addition of oligomycin, effectively shutting down oxidative phosphorylation and driving the cell to use glycolysis to its maximum capacity.

• Glycolytic reserve: This measure indicates the capability of a cell to respond to an energetic demand as well as how close the glycolytic function is to the cell’s theoretical maximum.

• Nonglycolytic acidification (basal levels): This measures other sources of extracellular acidification that are not attributed to glycolysis

We have added these details to materials & methods for the reader's understanding.

3. In Figure 3D, TGFβ group and TGFβ+PX-478 group had no significant increase of ECAR after giving glucose. Please explain it.

- In Figure 3D, TGF-β1 significantly upregulated the level of ECAR, and in the TGFβ+PX-478 group, glucose-induced ECAR upregulation was suppressed. This means that PX-478 inhibits cell reactivity by Glucose, that is, glycolysis. We demonstrated in Fig. 3 that PX-478 inhibits glycolysis-mediated factors HIF-1a, HK2, PFKFB3 and PFKFB4. Therefore, the level of ECAR did not increase significantly despite the administration of the substrate glucose.

 * Glucose administered at 15 min time-dependently increased the level of ECAR. (Red Graph)

4. The authors stated that they studied the role of glycolytic reprogram in CRS. However, they only focused on glycolysis, and provided no data regarding oxidative phosphorylation.

- Thanks for your comment. We used the XF substrate oxidation stress test kit to confirm the oxidative phosphorylation reaction and attach the relevant data as a supplement. This test method can measure the oxidative phosphorylation reaction to substrates such as glucose and pyruvic acid by evaluating the change in oxygen consumption of living cells. As a result, TGF-β1 increased the overall level of OCR, and PX-478 partially suppressed it.

  

5. The authors should compare the difference between CRSsNP and CRSwNP.

- CRS is divided into CRS without nasal polyp (CRSsNP) and CRS with nasal polyp (CRSwNP) according to the presence or absence of nasal polyp. It was reported that extracellular matrix deposition, fibrosis, massive immune cell infiltration were observed in the stroma of CRSsNP, and fibrin deposition, edema were observed in CRSwNP. The original intention of the study was to classify CRS into only CRSsNP and CRSwNP to see the difference in glycolytic reprogramming of fibroblasts. In fact, CRS is classified into Th1, Th2, and Th 17 types according to the immune response, and subclassified into eosinophilic and non-eosinophilic according to the infiltrated immune cells, so it was difficult to find a tendency in classification based on nasal polyp. Therefore, there was a limit to the detailed classification of the recruited patients. So the study was conducted based on CRS and normal subjects. In the future, we plan to subdivide the patient group according to the immune response and conduct research.

6. Chen et al recently studied glycolysis in CRS epithelial cells and its involvement in the pathogenesis of CRS (https://doi.org/10.1016/j.jaci.2022.09.036). This recent study should be compared and discussed.

- Thanks for your comment. We reviewed the papers you mentioned and additionally described their relevance to our research in the discussion section. Thank you.

Minor comment:

1. On line 354, the format of the reference was different from the others.

- We revised the reference style.

 

Reviewer #2

1. Did the authors study any cytokines both proinflammatory and anti-inflammatory.

- Thank you for your nice comments. As a follow-up study, we are conducting a study on the correlation between glycolytic reprogramming and proinflammatory cytokines, IL-6 and IL-8. In previous studies, we demonstrated that environmental hazardous substances such as Asian sand dust and diesel exhaust particles stimulate nasal fibroblasts to secrete proinflammatory cytokines such as IL-6 and IL-8 [1, 2]. Therefore, in the future, we plan to conduct a study on the role of glycolysis in the process of producing IL-6 and IL-8 by these external antigens.

2. In ECAR experiment did the authors had done normalization of data. If not please include in revised manuscript

- Thank you for your comment. ECAR experiments are conducted in two steps. First, equilibration is performed through a calibrant utility plate, and then the sensor cartridge is loaded to measure drug treatment and result values. Experimental data was normalized using Wave 2.6 software provided by Agilent. Normalized data were included in Figure 1A and 3D. All values shown in the graph are normalized values. Additionally, cell count was performed before measuring ECAR values, and through this, ECAR values were normalized. Relevant information will be attached to the manuscript.

3. The figures are not clear enough to read. Please include high quality figures with font size that is clearly visible and readable.

- Thank you for your comment. We replaced all figures as high quality figures. 

References

1. Yang, H.-W., J.-H. Park, J.-M. Shin, H.-M. Lee, and I.-H. Park, Asian Sand Dust Upregulates IL-6 and IL-8 via ROS, JNK, ERK, and CREB Signaling in Human Nasal Fibroblasts. American Journal of Rhinology & Allergy, 2019. 34(2): p. 249-261.

2. Kim, J.A., J.H. Cho, I.-H. Park, J.-M. Shin, S.-A. Lee, and H.-M. Lee, Diesel Exhaust Particles Upregulate Interleukins IL-6 and IL-8 in Nasal Fibroblasts. PLOS ONE, 2016. 11(6): p. e0157058.

---

## [Decision Letter · Decision Letter 1]

30 Jan 2023

Glycolytic reprogramming is involved in tissue remodeling on chronic rhinosinusitis

PONE-D-22-33119R1

Dear Dr. Shin,

We’re pleased to inform you that your manuscript has been judged scientifically suitable for publication and will be formally accepted for publication once it meets all outstanding technical requirements.

Kind regards,

Kishor Pant

Academic Editor

PLOS ONE

Reviewers' comments:

Reviewer's Responses to Questions

**Comments to the Author**

1. If the authors have adequately addressed your comments raised in a previous round of review and you feel that this manuscript is now acceptable for publication, you may indicate that here to bypass the “Comments to the Author” section, enter your conflict of interest statement in the “Confidential to Editor” section, and submit your "Accept" recommendation.

Reviewer #1: All comments have been addressed

Reviewer #2: All comments have been addressed

2. Is the manuscript technically sound, and do the data support the conclusions?

Reviewer #1: (No Response)

Reviewer #2: Yes

3. Has the statistical analysis been performed appropriately and rigorously? 

Reviewer #1: (No Response)

Reviewer #2: Yes

4. Have the authors made all data underlying the findings in their manuscript fully available?

Reviewer #1: (No Response)

Reviewer #2: Yes

5. Is the manuscript presented in an intelligible fashion and written in standard English?

Reviewer #1: (No Response)

Reviewer #2: Yes

6. Review Comments to the Author

Reviewer #1: (No Response)

Reviewer #2: The authors have addressed all the comments raised in the previous review. The manuscript should be accepted.

7. PLOS authors have the option to publish the peer review history of their article (what does this mean?). If published, this will include your full peer review and any attached files.

Reviewer #1: No

Reviewer #2: **Yes**

<quillbot-extension-portal></quillbot-extension-portal>

---

## [Editor Report · Acceptance letter]

7 Feb 2023

PONE-D-22-33119R1 

Glycolytic reprogramming is involved in tissue remodeling on chronic rhinosinusitis 

Dear Dr. Shin:

I'm pleased to inform you that your manuscript has been deemed suitable for publication in PLOS ONE. Congratulations! Your manuscript is now with our production department. 

Kind regards, 

on behalf of

Dr. Kishor Pant 

Academic Editor

PLOS ONE